# PolyViT: Co-training Vision Transformers on Images, Videos and Audio

**Valerii Likhosherstov**[*][†]                                                                *vl304@cam.ac.uk*
*University of Cambridge*

**Anurag Arnab**[*]                                                                            *aarnab@google.com*
*Google Research*

**Krzysztof Choromanski**                                                                      *kchoro@google.com*
*Google Research*

**Mario Lucic**                                                                                *lucic@google.com*
*Google Research*

**Yi Tay**                                                                                     *yitay@google.com*
*Google Research*

**Adrian Weller**                                                                              *aw665@cam.ac.uk*
*University of Cambridge & The Alan Turing Institute*

**Mostafa Dehghani**[*]                                                                        *dehghani@google.com*
*Google Research*

**Reviewed on OpenReview:** *https://openreview.net/forum?id=zKnqZeUCLO*

## Abstract

Can we train a single transformer model capable of processing multiple modalities and datasets, whilst sharing almost all of its learnable parameters? We present PolyViT, a model trained on images, audio and video to answer this question. PolyViT consists of a single transformer backbone, modality-specific tokenizers and task-specific output heads. By co-training on different tasks of a single modality, we are able to achieve significant accuracy improvements on 5 standard video- and audio-classification datasets. Furthermore, co-training PolyViT on multiple modalities and tasks leads to a parameter-efficient model which generalizes across multiple domains. In particular, our multi-modal PolyViT trained on 9 datasets across 3 modalities uses 8.3 times fewer parameters and outperforms a state-of-the-art single-task baseline on 2 of these datasets, whilst achieving competitive performance on the others. Finally, this simple and practical approach necessitates less hyperparameter tuning as the per-task hyperparameters can be readily reused. To facilitate further research, we have released code at https://github.com/google-research/scenic.

## 1 Introduction

Transformers (Vaswani et al., 2017) are a flexible family of neural sequence-to-sequence models. While they were originally designed for natural language processing, they have recently been adapted to a range of perception tasks, such as classification of images (Dosovitskiy et al., 2021), video (Arnab et al., 2021) and audio (Gong et al., 2021). Despite recent advances across different domains and tasks, current state-of-the-art methods train a separate model with different model parameters for each task at hand.

---

[*]Equal contribution.
[†]The work was done during author's internship at Google.

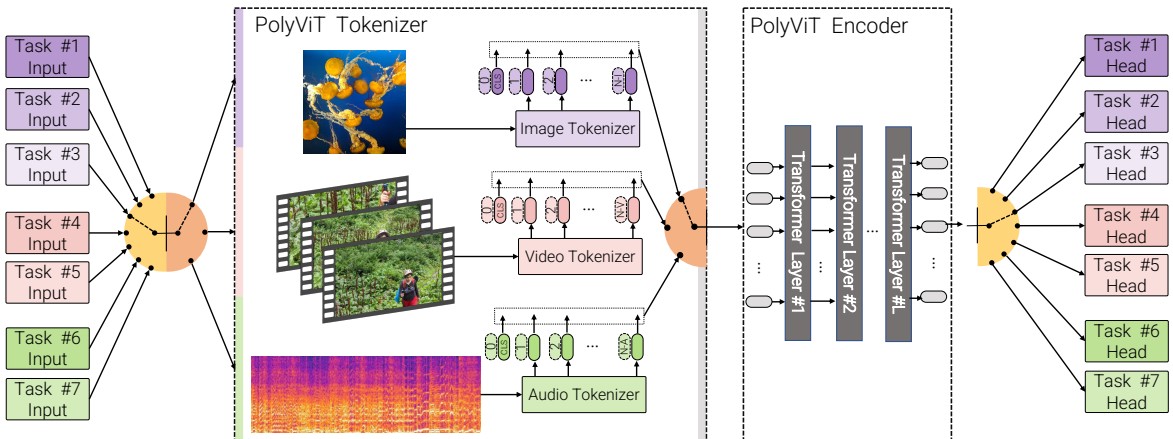

Figure 1: Overview of PolyViT. Our model is capable of performing multiple tasks spanning different modalities, and processes a single task at a time. The architecture consists of a transformer encoder shared among all tasks, modality-specific input tokenizers and task-specific output heads.

In this work, we present a simple yet effective method of training a single, unified model (Fig. 1) that achieves competitive, or state-of-the-art results for image-, video-, and audio-classification. We go beyond using a common architecture for different modalities (Jaegle et al., 2021), as we also share model parameters across tasks and modalities, thus enabling potential synergies. Our approach is motivated both technically, by the fact that transformers are generic architectures that can operate on any modality that can be tokenized, and intuitively, since human perception is inherently multimodal and performed by a single brain.

Our main technique is *co-training*: training a single model on multiple classification tasks (across potentially multiple modalities) simultaneously. We consider various settings, and simultaneously solve as many as 9 different image-, video- and audio-classification tasks. Our model is capable of performing multiple tasks, but performs a single task at a time for a given input (see Fig. 1).

Although similar techniques have been explored in computer vision (Maninis et al., 2019) and natural language processing (Raffel et al., 2019), to our knowledge, this is the first approach using multiple modalities and achieving state-of-the-art performance.

The proposed co-training approach has several benefits: It is parameter-efficient as we share the transformer parameters for each of the $n$ tasks of interest, approximately reducing the number of parameters by a factor of $n$. This has practical advantages when deploying models on computationally constrained devices (such as smartphones) with limited memory which may not otherwise be able to fit the weights of $n$ different models (Iandola et al., 2017; Dehghani et al., 2021a). Moreover, maintaining a single model for multiple tasks simplifies model deployment and online updates (Iandola et al., 2017). Furthermore, co-training on tasks of the same modality leads to accuracy improvements on each individual task whilst also linearly decreasing total parameters. In particular, we achieve significant improvements on video and audio classification across 5 different datasets. This is facilitated by our observation that co-training has a regularizing effect, that improves performance on smaller datasets that large transformer models would otherwise overfit on. In addition, when we extend co-training to multiple tasks and modalities, we observe that our accuracy is still competitive with the state-of-the-art whilst being even more parameter-efficient – our model trained on 9 datasets uses 8.3 times fewer parameters whilst having at most a 1.2% accuracy drop compared to state-of-the-art single-task baselines. Finally, linear probing experiments show that this multi-task, multi-modal model is able to learn representations that generalize across multiple tasks and domains. Once again, this has practical advantages when deploying models, as it shows that we can add new capabilities to the model by simply training an additional linear classifier.

In addition to all the benefits outlined above, our co-training setup is simple and practical to implement. It does not require hyperparameter tuning for each combination of co-training datasets, as we can readily

adapt the settings of standard, single-task training. In addition, co-training does not increase the overall training cost either, as the total number of training steps does not exceed that of the sum of each single-task baseline.

## 2 Related Work

Our model is related to multi-task learning and transformer models, which we discuss below.

**Multi-task learning**  Multi-task learning aims to develop models that can address multiple tasks whilst sharing parameters and computation between them (Caruana, 1997). In computer vision, multiple papers have developed models which predict multiple outputs (for example semantic segmentation and surface normals), given a single input image (Eigen & Fergus, 2015; Kokkinos, 2017; Zhang et al., 2014). Numerous works have also observed that although multi-task models are more versatile, their accuracies are lower than single-task models, and this accuracy deficit increases with the number of tasks, or by simultaneously performing unrelated tasks (Kokkinos, 2017; Zamir et al., 2018; McCann et al., 2018). Moreover, jointly training a network to simultaneously perform multiple tasks has typically required careful calibration of the individual tasks, to ensure that none of the task-specific losses dominates another. Methods to mitigate this include gradient-normalization (Chen et al., 2018) and -surgery (Yu et al., 2020) and adaptive loss weights (Sener & Koltun, 2018; Kendall et al., 2018) and curriculums (Guo et al., 2018) among others.

Our work differs in that although our network is capable of performing multiple tasks, it performs one task at a time for a given input. Note that this setting is also more suited to the case of handling multiple input modalities. Such an approach was also taken by (Maninis et al., 2019) who named it "single-tasking of multiple tasks" in the context of computer vision. However, in natural language processing (NLP), this setting is still referred to as "multi-task learning" (Collobert & Weston, 2008). Furthermore, our co-training strategy is simple, and alternates between performing SGD for batches of separate tasks. For high-capacity transformer models, we find that co-training on multiple datasets simultaneously helps to regularize the model on a dataset that it would otherwise overfit on, thus achieving accuracy improvements from co-training. Previous works have improved performance on additional tasks only by introducing extra task-specific parameters (Misra et al., 2016; Houlsby et al., 2019) which are typically conditioned on the input (Rebuffi et al., 2017; Maninis et al., 2019).

We also note that similar co-training setups to our work have been explored in NLP. A recent paradigm in NLP has been to reduce different tasks to a common, unified framework (Raffel et al., 2019; Brown et al., 2020; McCann et al., 2018). This common interface allows co-training a single model to perform multiple tasks, as it effectively involves concatenating multiple datasets together (Raffel et al., 2019; Khashabi et al., 2020; Tay et al., 2020).

Although the majority of previous multi-task learning works have considered only a single modality, (Kaiser et al., 2017) presented an early effort on multi-modal models. Their propose a heterogeneous model consisted of convolutional layers to process images, and attention and mixture-of-experts layers to model text. This work, however, did not analyse how to co-train these different tasks and modalities as our paper. Most recently, proposed using multi-task learning as the mean for transfer learning and showed that co-training upstream and downstream tasks, we can improve the transfer results compared to traditional pretraining-finetuning as sequential processes.

Most recently, Arnab et al. (2022) proposed using multi-task learning as a means for transfer learning and demonstrated that by co-training upstream and downstream tasks, the transfer results can be improved compared to traditional pretraining-finetuning as sequential processes.

**Transformer models**  Our model, motivated by (Dosovitskiy et al., 2021), can readily handle diverse modalities, as transformers operate on any sequence of tokens. Relevant to us, Perceiver (Jaegle et al., 2021) and Perceiver-IO (Jaegle et al., 2022) is a transformer architecture that can process different modalities. Instead of tokenizing images or audio spectrograms with non-overlapping patches like (Dosovitskiy et al., 2021) and (Gong et al., 2021) respectively, (Jaegle et al., 2021) operate directly on the raw input by projecting it into a smaller, latent set of tokens using cross-attention. Although this architecture is capable of processing

different modalities, the authors train separate networks with separate parameters for each task. Therefore, they do not consider co-training scenarios like our work. MBT (Nagrani et al., 2021), on the other hand, proposes a transformer model to fuse different modalities (for example audio and rgb frames of videos) to solve a single task. Once again, separate model parameters are used for each task.

UniT (Hu & Singh, 2021) co-train a transformer-based model, but specifically for vision-and-language tasks. The authors use an encoder-decoder architecture (Vaswani et al., 2017), where only the decoder is shared among different tasks, and the encoder is specialized for each modality. In particular, the visual encoder is DeTR (Carion et al., 2020) and the text encoder is BERT (Devlin et al., 2019), and each component is pretrained separately. In contrast to our work, they do not consider scenarios where the entire transformer backbone is shared among different tasks, nor do they thoroughly analyze how to co-train multiple tasks and modalities like our work. Furthermore, their approach does not outperform single-task baselines as our work does. Other papers concentrating on multi-task learning of vision-and-language tasks include (Lu et al., 2019; Li et al., 2020; Lu et al., 2020). On a separate track, (Bain et al., 2021) use a single transformer encoder to process both images and videos for video-text retrieval. However, their model is still trained on a single dataset and task, and the authors process images with the transformer as part of a complex, curriculum-learning strategy. This is in contrast with our work which simultaneously trains a model for multiple tasks across different modalities.

Additionally, we note that transformers have been used to process multiple modalities (Akbari et al., 2021; Lee et al., 2021) for cross-modal self-supervised learning (Alayrac et al., 2020; Miech et al., 2020). (Lee et al., 2021) train a transformer on top of visual and audio features obtained from convolutional networks. And to make model training computationally feasible, they perform low-rank approximations of the parameter matrices (Sainath et al., 2013; Yang et al., 2015) to reduce the total number of model parameters. These approaches are thus complementary to our work which shares almost all parameters in the model among different tasks.

Finally, we note that a concurrent work, Unified-IO (Lu et al., 2022) proposes a sequence-to-sequence model which can can perform various vision and language tasks with the same model. Although this work can handle a wide range of tasks, it is however, not competitive with the state-of-the-art as our approach.

## 3 Preliminaries

We define a *modality* as the type of input processed by the network. In this work, we consider images, audio, and video (specifically, the sequence of image frames in a video) as three separate modalities. We perform classification as it is a fundamental problem whose solutions are often extended to more complex ones (Girshick et al., 2014; He et al., 2017). By *task*, we refer to a pair of input modality and a set of classes from which one or multiple classes are to be selected for a given input. Each task corresponds directly to a dataset, for example, ImageNet-1K (Deng et al., 2009) for image classification or Kinetics 400 (Kay et al., 2017) for video classification.

### 3.1 Vision Transformers and extensions

The Vision Transformer (ViT, (Dosovitskiy et al., 2021)) is a transformer-based architecture for image classification that closely follows Vaswani et al. (2017). In contrast to language which is intuitively tokenized into words, ViT extracts tokens from the input image, $\mathbf{x}^{\text{IMG}} \in \mathbb{R}^{H \times W \times 3}$, by splitting it into $N = \lfloor H/h \rfloor \times \lfloor W/w \rfloor$ non-overlapping patches, $\mathbf{x}_1, \ldots, \mathbf{x}_N \in \mathbb{R}^{h \times w \times 3}$. Each patch, $x_i$, is then projected into a token $\mathbf{z}_i \in \mathbb{R}^d$ by a linear operator $\mathbf{E}$, $\mathbf{z}_i = \mathbf{E}\mathbf{x}_i$ (*input embedding operator*). All tokens are then concatenated into a sequence, which is also prepended with a learnable *class token* $\mathbf{z}_{cls} \in \mathbb{R}^d$. Learnable *positional embeddings* $\mathbf{p} \in \mathbb{R}^{(N+1) \times d}$ are also added to this sequence as the transformer is otherwise permutation invariant. We denote this tokenization process as

$$\mathbf{z}^0 = \begin{bmatrix} \mathbf{z}_{cls} & \mathbf{E}\mathbf{x}_1 & \ldots & \mathbf{E}\mathbf{x}_N \end{bmatrix} + \mathbf{p}. \tag{1}$$

Note that the linear operator $\mathbf{E}$ can also be seen as a 2D convolution with kernel of size $h \times w$ and strides $(h, w)$. The sequence of tokens, $\mathbf{z}$, is then processed by a transformer encoder, consisting of $L$ layers. Each

layer, $\ell$, is applied sequentially, and performs the transformations,

$$\mathbf{y}^\ell = \text{MSA}\left(\text{LN}\left(\mathbf{z}^{\ell-1}\right)\right) + \mathbf{z}^{\ell-1} \tag{2}$$

$$\mathbf{z}^\ell = \text{MLP}\left(\text{LN}\left(\mathbf{y}^\ell\right)\right) + \mathbf{y}^\ell, \tag{3}$$

where MSA denotes multi-head self-attention (Vaswani et al., 2017), MLP is a neural network with a single hidden layer and a GeLU nonlinearity (Hendrycks & Gimpel, 2016), and LN denotes layer normalization (Ba et al., 2016).

For a $C$-class classification problem, the class logits produced by the model are obtained by applying an output *linear head* on the encoded classification token, $\mathbf{z}_{cls}^L$, as

$$\mathbf{W}_{out}\mathbf{z}_{cls}^L + \mathbf{b}_{out} \in \mathbb{R}^C, \tag{4}$$

where $\mathbf{W}_{out} \in \mathbb{R}^{C \times d}$ and $\mathbf{b}_{out} \in \mathbb{R}^C$ are the linear head's learnable parameters.

**Extensions of ViT to audio and video** The *Audio Spectrogram Transformer* (AST) (Gong et al., 2021) follows the same architecture as ViT, with the only difference that its inputs are log-mel spectrograms. Spectrograms are image-like, time-frequency representations of audio, and can be tokenized like images. Moreover, the best AST model was initialized from ViT models pretrained on large image datasets.

*Video Vision Transformers* (ViViT) (Arnab et al., 2021) are an extension of ViT to video. The authors proposed four model variants, and we consider the unfactorized one (Model 1 in (Arnab et al., 2021)). This model differs from ViT in the input tokenization process, which it extends from 2D image patches to 3D spatio-temporal "tubelets". Namely, a video input $\mathbf{x}^{\text{VID}} \in \mathbb{R}^{F \times H \times W \times 3}$ is split into $N = \lfloor F/f \rfloor \times \lfloor H/h \rfloor \times \lfloor W/w \rfloor$ non-overlapping tubelets $\mathbf{x}_1, \ldots, \mathbf{x}_N \in \mathbb{R}^{f \times h \times w \times 3}$. Following ViT, a linear operator $\mathbf{E}^{\text{VID}}$, which can be interpreted as a 3D convolution, projects $\{\mathbf{x}_i\}$ into a sequence of tokens $\{z_i = \mathbf{E}^{\text{VID}}\mathbf{x}_i \in \mathbb{R}^d\}$, and repeats computations (1-4).

**Initialization** Finally, note that ViT, ViViT and AST all achieve their highest performance when pretrained on a large-scale dataset such as ImageNet-21K (Deng et al., 2009) or JFT (Sun et al., 2017). More specifically, ViT was initially pretrained on ImageNet-21K or JFT, and then finetuned at higher resolution on target datasets such as ImageNet-1K. ViViT and AST also initialize their models from large-scale, image-pretrained models (Abnar et al., 2021). In all of these cases, the positional embeddings, $\mathbf{p}$, which depend on the sequence length $N$ (and thus the input resolution), are interpolated from the pretrained model to the finetuned model. Furthermore, the 3D embedding projection of ViViT, $\mathbf{E}^{\text{VID}}$, is initialized from the 2D projection of ViT, $\mathbf{E}^{\text{IMG}}$ (Arnab et al., 2021).

The similarities between ViT, ViViT and AST allow us to construct a multi-modal model with a shared transformer encoder, and separate input tokenizers as described next.

inference

# 4   Co-training ViT on images, audio and video

## 4.1   PolyViT architecture

PolyViT is a single architecture that is capable of processing inputs from multiple modalities. As shown in Fig. 1, we share a transformer encoder among different tasks and modalities, enabling up to a linear reduction in parameters with the number of tasks. Note that PolyViT with $L$ layers acts like an $L$-layer ViT when processing images, an $L$-layer AST when processing audio, and an $L$-layer unfactorized ViViT when processing video. While capable of handling multiple modalities, it performs one task from one modality in a given forward pass.

As shown in Fig. 1, PolyViT employs modality-specific class tokens, $\mathbf{z}_{cls}^{\text{IMG}}$, $\mathbf{z}_{cls}^{\text{VID}}$, $\mathbf{z}_{cls}^{\text{AUD}}$, input embedding operators, $\mathbf{E}^{\text{IMG}}, \mathbf{E}^{\text{VID}}, \mathbf{E}^{\text{AUD}}$, and positional embeddings $\mathbf{p}^{\text{IMG}}$, $\mathbf{p}^{\text{VID}}$, $\mathbf{p}^{\text{AUD}}$. This allows the network to encode modality-specific information that can be leveraged by the subsequent, shared transformer backbone. It also accounts for the fact that the number of tokens per modality may vary.

A separate output linear head (Eq. 4) is then used for each task with the following learnable weights:

$$\mathbf{W}_{out} = \mathbf{W}_{out}^{mod,j} \in \mathbb{R}^{C_j \times d}, \; \mathbf{b}_{out} = \mathbf{b}_{out}^{mod,j} \in \mathbb{R}^{C^j}, \tag{5}$$

where $mod \in \{\text{IMG}, \text{VID}, \text{AUD}\}$ is a modality, $j \in \{1, \ldots, T^{mod}\}$ is a task index in the set of tasks for that modality, $T^{mod}$ is the number of tasks for that modality and $C_j$ is the number of classes for that task. Note that the output heads are the only task-specific parameters. The input embedding operators, positional embeddings and class tokens are shared by all tasks within a modality.

To increase model capacity when co-training on a large number of tasks and modalities simultaneously, we can optionally include $L_{adapt} \geq 0$ modality-specific transformer layers (which we denote as *modality-adaptor layers*). These transformer layers are applied directly after tokenization. In this case, there are $L_{shared} = L - L_{adapt}$ layers which are shared among all modalities and tasks. We can think of this case as using a shallower transformer encoder, but a deeper subnetwork to extract tokens from each modality.

As almost all computation and parameters within our architecture are within the $L$ layers of the transformer encoder, if there are $n$ tasks, we reduce the total number of parameters by a factor of approximately $n$ when $L_{shared} = L$. This is in comparison to standard, single-task training. Note that the overall inference time does not change, as PolyViT still performs one task per forward pass.

## 4.2 Co-training procedure

We optimize all PolyViT model parameters, $\theta$, simultaneously across all the tasks that we are co-training with first-order gradient-based optimizers such as stochastic gradient descent (SGD). For brevity, we will refer to each iteration of one of these optimizers as an "SGD step" below. There are a myriad of design choices on how to construct training batches, compute gradients, and tune the training hyperparameters.

In all cases, we construct our training minibatches using examples from a single task. This design choice allows us to evaluate gradients and perform a parameter update using the same training hyperparameters (e.g., learning rate, batch size, and momentum) as a conventional single-task baseline. As a result, we can perform co-training on multiple tasks without any additional hyperparameter tuning compared to the single-task baseline (Dosovitskiy et al., 2021; Gong et al., 2021; Arnab et al., 2021), making co-training simple to perform in practice, and alleviating the need to perform large hyperparameter sweeps in order to achieve competitive accuracy. Note that without this property, we would need to tune training hyperparameters on the product set of all co-training datasets, which would be computationally infeasible. Constructing minibatches from a single task (where each example has the same number of tokens) has further computational advantages on GPU- or TPU-accelerators, as tokens do not need to be padded to a maximum sequence length.

During co-training, for each SGD step, we sample a task (dataset), then sample a minibatch from that task, evaluate a gradient and then perform a parameter update. An important consideration is the order in which we sample tasks and whether we accumulate gradients over different minibatches and tasks. We describe several task sampling schedules below and in Fig. 2. We first denote $U_j$ as the number of SGD steps for the single-task baseline that the original authors reported for their best model, where $j \in \{1, \ldots, T\}$ indexes the task and $T = T^{\text{IMG}} + T^{\text{AUD}} + T^{\text{VID}}$ is the total number of tasks. Furthermore, we define $U$ as the total number of SGD steps during co-training.

**Task-by-task**   In this schedule, the first $U_{j_1}$ SGD steps are performed with task $j_1$, the next $U_{j_2}$ steps using task $j_2$ and so on, where $[j_1, \ldots, j_T]$ is a random task order.

**Alternating**   This deterministic schedule alternates between tasks in a fixed, repeating order. Concretely, we perform a single SGD step for each task in sequence before repeating the same order. We set $U = \sum_{j=1}^{M} U_j$ which implies $U/T$ training steps per task.

**Uniform task sampling**   This is a stochastic version of the schedule above, where the task for each SGD step is sampled from a uniform distribution, with probability $1/T$. We implement it such that the number

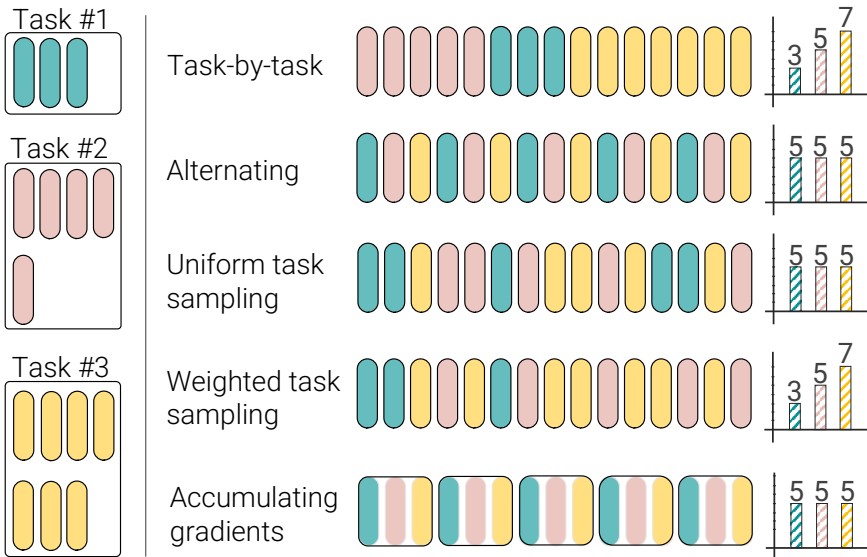

Figure 2: Task sampling schedules. Each element within a task corresponds to the number of training steps performed for that task by the baseline model.

of training steps for task $j$ is exactly $U/T$, by randomly permuting an array with $U$ elements, where $U/T$ elements correspond to each task.

**Weighted task sampling** In this stochastic schedule, we sample each task with a weight proportional to the number of training steps in the single-task baseline. Therefore, $U = \sum_{j=1}^{M} U_j$, and the sampling weight for task $j$ is $U_j/U$. We implement this schedule as above, to ensure that we perform exactly $U_j$ steps for task $j$.

**Accumulating gradients** For $T$ tasks, we perform a forward and backward pass on a minibatch for each task, summing the gradients over each task. We then perform a single parameter update with the average of the accumulated gradients, thus effectively using a larger batch size encompassing all the tasks being co-trained. Here, we set $U = (\sum_{j=1}^{T} U_j)/T$.

## 4.3 Initialization of PolyViT

As described in Sec. 3.1, ViT, ViViT and AST models are initialized from models pretrained on ImageNet-21K or JFT before being finetuned for the task of interest. In our experiments, we also finetune from a ViT model pretrained on ImageNet-21K, and follow the initialization methods for the positional embeddings, **p**, and input embeddings, **E**, for each modality as described in Dosovitskiy et al. (2021) and Arnab et al. (2021).

When we use modality-adaptor layers, that is $L_{adapt} > 0$, the first $L_{adapt}$ layers for each modality are initialized with the same first $L_{adapt}$ layers from the pretrained ViT model. These parameters are however allowed to change from each other during training. Similarly, shared PolyViT layers are initialized from the last $L_{shared}$ transformer encoder layers from the pretrained ViT model. Note that the output linear heads are all initialized randomly.

Table 1: The effect of the task sampling schedule on co-training performance on multiple modalities and tasks. The highest accuracy is shown in bold, the second-highest is underlined. The "Weighted" task sampling method consistently achieves the highest accuracy for 8 out of 9 tasks, and second-highest on the remainder. Results are on the validation set. Color bar: (lowest) ▮▬▬▬▬ (highest).

| | Image | | | | | Video | | Audio | |
|---|---|---|---|---|---|---|---|---|---|
| Schedule | Im1K | C100 | C10 | Pets | R45 | K400 | MiT | MiniAS | VGG |
| Task-by-task | 0.3 | 0.8 | 11.7 | 1.9 | 2.0 | 0.3 | 0.3 | 1.6 | 37.2 |
| Accumulated | **88.1** | 90.0 | 98.8 | 94.0 | 96.1 | 58.0 | 22.5 | 22.9 | 27.3 |
| Alternating | 86.0 | 89.4 | 99.2 | 94.0 | 95.8 | 69.7 | 30.0 | 31.4 | 44.6 |
| Uniform | 85.8 | 89.3 | 98.6 | 94.6 | 96.1 | 68.8 | 29.3 | 30.6 | 44.1 |
| Weighted | 86.9 | **90.4** | **99.3** | **96.5** | **97.0** | **71.6** | **32.5** | **33.5** | **49.2** |

# 5 Experiments

## 5.1 Experimental Setup

We train PolyViT simultaneously on 9 diverse classification tasks spanning the image, video, and audio modalities. Note that datasets and tasks have a one-to-one correspondence. We chose this setup of 9 tasks, as the datasets include a large variation in domain and training set sizes. Furthermore, the single-task baseline training hyperparameters vary substantially between the tasks. Consequently, we believe this presents a challenging co-training setup.

When co-training for image classification, we use ImageNet-1K, CIFAR-10 and -100, Oxford-IIIT Pets, and RESISC45. For video, we use Kinetics 400 and Moments in Time, and for audio, AudioSet and VGGSound. Exhaustive details of these datasets are in Appendix A. As in Nagrani et al. (2021), we evaluate on the whole AudioSet validation set, but use a smaller balanced subset referred to Mini-AudioSet (MiniAS) for initial experiments. We then use a larger, balanced subset of 500 000 examples (referred to AS-500k) for our state-of-the-art comparisons following Nagrani et al. (2021). We follow standard evaluation protocols for each task, reporting classification accuracy (%) for all tasks except AudioSet, where we report mean average precision (mAP) as it is a multilabel dataset.

We set the training hyperparameters for these tasks (and those of the single-task baselines) using the values reported by Dosovitskiy et al. (2021) for image tasks, Arnab et al. (2021) for video tasks and Nagrani et al. (2021) for audio tasks (detailed in Appendix A). Note that the "audio-only" model of Nagrani et al. (2021), which we use as our baseline, is identical to AST Gong et al. (2021), and we choose it since the authors have evaluated on more datasets.

We perform experiments with two standard transformer encoder configurations: Base (number of layers, $L = 12$, hidden dimension $d = 768$, attention heads $h = 12$) and Large ($L = 24$, $d = 1024$, $h = 16$) following Devlin et al. (2019); Dosovitskiy et al. (2021). As in Dosovitskiy et al. (2021), we initialize our PolyViT model and baselines with ViT pretrained on ImageNet-21K. We refer to this initialized model as ViT-Im21K. PolyViT is implemented in Scenic (Dehghani et al., 2021b) and the code for training and evaluation of the model is available at https://github.com/google-research/scenic.

## 5.2 Selecting the best task sampling schedule for co-training

We begin by analyzing the effect of the different task sampling schedules listed in Sec. 4.2. We use the full, aforementioned 9-task set-up with PolyViT-Base and all encoder layers shared ($L_{shared} = L = 12$, $L_{adapt} = 0$).

As shown in Tab. 1, the "Task-by-task" schedule performs poorly, and only achieves decent performance on one task, as it suffers from catastrophic forgetting (French, 1999). The "Accumulated" sampling strategy requires using a single learning rate for all tasks (since the accumulated gradient over all tasks is used for

Table 2: Co-training with PolyViT-Base. As indicated by the "#Models" column, some rows correspond to multiple trained models. In this case, we report the total number of parameters across all models. PolyViT co-trained on a single-modality outperforms single-task baselines in most cases, whereas PolyViT co-trained on multiple modalities achieves competitive performance with a large reduction in parameters (test set results). Color bar: (lowest) �\[\] (highest). Further details in Appendix A.

| Model | #Models | #Params | Image | | | | | Video | | Audio | |
| | | | Im1K | C100 | C10 | Pets | R45 | K400 | MiT | MiniAS | VGG |
|---|---|---|---|---|---|---|---|---|---|---|---|
| ViT-Im21K Linear probe | 1 | **93M** | 80.7 | 76.2 | 91.7 | 91.8 | 81.7 | 64.0 | 25.5 | 11.3 | 15.7 |
| Single-task baseline | 9 | 773M | 83.1 | 92.0 | 99.0 | 94.5 | **96.7** | 78.7 | 33.8 | 29.3 | **51.7** |
| PolyViT, 1 modality | 3 | 263M | **84.3** | **93.3** | **99.1** | **95.1** | 96.4 | **80.2** | **36.5** | **36.7** | 51.6 |
| PolyViT, $L_{adapt} = 0$ | 1 | **93M** | 83.1 | 91.2 | 99.0 | 95.0 | **96.7** | 77.5 | 33.2 | 32.3 | 50.6 |
| PolyViT, $L_{adapt} = L/2$ | 1 | 178M | 82.8 | 91.5 | 99.0 | 95.0 | 96.6 | 79.4 | 35.3 | 33.1 | 51.5 |

performing a parameter update). As we used a learning rate of 0.03, which is the learning rate used by the image tasks, and significantly lower than the learning rates for the video and audio tasks of the baselines (details in Appendix A), this method only performs well on image datasets. The "Alternating", "Uniform" and "Weighted" strategies perform the best, showing that task-specific learning rates, and switching between gradient-updates for different tasks is crucial for accuracy.

In particular, the "Weighted" sampling method performs the best, achieving the highest accuracies on 8 of the 9 tasks (and second-highest on the remainder), motivating us to use it for all subsequent experiments. We postulate that this is the case, because unlike the "Uniform" and "Alternating" strategies, the "Weighted" strategy performs a different number of update steps for each task. For example, small datasets with a lower number of training steps in their baseline training configurations are sampled less frequently, which can help to mitigate overfitting. An example is the Pets task, that is only sampled for 500 iterations, out of the 417 000 total steps, or just 0.11% of the SGD updates. Another advantage of the "Weighted" strategy is that it performs the same number of steps per task as a single-task baseline. Therefore, it uses the same computational resources during training as 9 separate, single-task baselines. Our experiment also shows that if we do not have training hyperparameters for a new task, we can simply tune them separately in the single-task setting, and then reuse them for co-training. This approach requires significantly less computation than tuning training hyperparameters directly in the co-training setup.

## 5.3 Co-training with PolyViT

Table 2 presents approaches for training models to solve 9 different tasks across the image, video and audio modalities. We consider two variants of PolyViT: The first is PolyViT for a single modality, where we co-train three separate PolyViT models on all the tasks from either the image, video or audio modalities. The second is the multi-modal PolyViT scenario where we co-train on all nine tasks across three modalities. Here, we set $L_{adapt}$ to 0 and $L/2$ respectively to understand the effect of the number of modality-adaptor and shared layers.

**Baselines.** We compare PolyViT to the following two baselines, which illustrate two alternatives to co-training.

*Single-task baseline* The first baseline is to train 9 separate single-task models for each dataset, either ViT, ViViT or AST depending on the modality. This results in accuracies comparable to the state-of-the-art on the respective datasets, but also the largest number of total parameters.

*ViT-Im21K Linear probe baseline* The second baseline is to use a ViT model initialized on ImageNet-21K (ViT-Im21K) and to "freeze" the encoder of the network and train only the linear output heads (Eq. 4,5) for each task. Positional embeddings, **p**, and input embeddings, **E** are initialized following the methods used by ViT, ViViT or AST as described in Sec. 3.1. This baseline has the same number of parameters as PolyViT with $L_{adapt} = 0$. We choose a model pretrained on ImageNet-21K (Im21K) as this is the largest, publicly available image dataset that we are aware of.

Table 3: Linear probing of PolyViT and single-task baselines. Similar to the protocol for evaluating self-supervised representation learning, we train only a linear classifier on top of a "frozen" transformer encoder. Note how PolyViT co-trained on all tasks transfers well to all other datasets and modalities. Models trained on audio do not transfer well to images and video, and vice versa. All models are pretrained on ImageNet-21K, and then optionally finetuned on downstream datasets. Color bar: (lowest) ▇▇▇▇▇▇ (highest).

| Model | Finetuning | Image | | | | | | Video | | | Audio | |
|---|---|---|---|---|---|---|---|---|---|---|---|---|
| | | C-ch101 | SUN397 | Dmlab | DTD | KITTI | PCAM | Epic K. | S-S v2 | K600 | MiT | K400 |
| ViT-Im21K pretrained | – | 88.9 | 75.7 | 41.0 | 72.1 | 46.9 | 80.2 | 10.0 | 17.8 | 66.6 | 4.9 | 10.8 |
| ViT | ImageNet-1K | **91.0** | 79.3 | 45.6 | 71.9 | 52.5 | 80.7 | 12.2 | 18.5 | 67.9 | 5.3 | 12.0 |
| PolyViT | Image tasks | 90.7 | **80.0** | 45.2 | **72.5** | 53.8 | 81.2 | 12.1 | 17.9 | 67.9 | 5.3 | 11.9 |
| ViViT | MiT | 85.2 | 73.8 | 43.0 | 69.9 | **54.9** | 81.7 | 14.9 | 26.3 | 74.2 | 5.1 | 11.9 |
| PolyViT | Video tasks | 89.2 | 77.5 | **45.9** | 71.1 | 53.5 | **83.8** | 17.2 | 27.9 | **79.7** | 5.3 | 12.2 |
| AST | VGGSound | 29.0 | 7.6 | 29.8 | 34.7 | 45.1 | 79.5 | 2.9 | 4.6 | 10.6 | 9.7 | 21.7 |
| PolyViT | Audio tasks | 38.8 | 14.7 | 31.4 | 40.1 | 43.2 | 78.4 | 3.0 | 5.8 | 14.5 | **10.3** | **22.0** |
| PolyViT $L_{adapt}=0$ | All | **91.0** | 78.2 | 45.8 | 71.8 | 52.3 | 81.9 | 16.8 | 27.9 | 77.8 | 9.6 | 20.6 |
| PolyViT $L_{adapt}=L/2$ | All | 90.7 | 77.8 | 45.1 | 72.1 | 52.5 | 82.3 | **18.0** | **28.7** | 79.4 | 9.9 | 21.1 |

**Discussion.** Table 2 shows that PolyViT trained on a single modality achieves the highest performance on 7 of the 9 datasets. On the remaining two, the accuracy difference is negligible, as it is at most 0.3%. Moreover, the total number of parameters is 3 times less than the single-task baselines. Single-modality co-training improves accuracy the most on the smaller datasets within the modality (Kinetics 400 in video, Mini-AudioSet for audio, and CIFAR-100 for images; full dataset details in Appendix A). This suggests that co-training acts as a regularizer, as noted by Caruana (1997), that facilitates learning on smaller datasets where high-capacity models would otherwise overfit.

Multi-modal PolyViT (final two rows) achieves competitive performance whilst using substantially fewer parameters. In particular, PolyViT with all transformer encoder layers shared between modalities ($L_{adapt} = 0$) is within 1.2% of the single-task baselines across all datasets whilst using 8.3 times fewer parameters. This model also comprehensively outperforms the ViT-Im21K Linear probe baseline which has the same number of parameters. Sharing half the transformer layers between modalities ($L_{adapt} = L/2$) increases model capacity, and the model improves upon the corresponding single-task baseline on 4 datasets, whilst being at most 0.5% worse on the others. The total number of parameters is still reduced by a factor of 4.3 compared to the single-task baselines.

Our results are consistent when using the Large model backbone as shown in Appendix C.

### 5.4 Evaluating learned representations with linear probes

We now evaluate the feature representations learned by PolyViT by simply appending and training only a new linear head (Eq. 4,5) for a new task. This evaluation therefore follows the experimental setting commonly used in self-supervised learning to evaluate the quality of learned representations (Chen et al., 2020; Grill et al., 2020). Note that the new task can come from any one of the three modalities that PolyViT is trained on, since the modality-adaptor layers (if present) are modality-specific rather than task-specific.

In particular, we evaluate on a number of new image, audio and video datasets as detailed in Appendix D. For image classification, we include Caltech101, SUN397, DmLab, DTD, Kitti Distance and PatchCamelyon, which are datasets from the Visual Task Adaptation Benchmark (Zhai et al., 2019) not in our co-training set. For video classification, we also include Epic Kitchens, Something-Something v2 and Kinetics 600. Finally, for audio classification, we use the audio versions of Moments in Time and Kinetics 400.

Table 4: Comparison to MBT Nagrani et al. (2021), the current published state-of-the-art using the same protocols. The second and third rows show that MBT, when first trained on AudioSet and then finetuned on VGGSound, and vice-versa, does not perform as well as PolyViT, showing that the regularizing benefits of co-training are not simply because the co-trained model has access to more data.

| Model | #Models | #Params | VGGSound Top 1 | Top 5 | AudioSet mAP |
|---|---|---|---|---|---|
| MBT (audio-only) | 2 | 172M | 52.3 | 78.1 | 44.3 |
| MBT: AS500k → VGGSound | 1 | 87M | 54.4 | **81.4** | 34.2 |
| MBT: VGGSound → AS500k | 1 | 87M | 22.1 | 43.5 | 44.4 |
| PolyViT | 1 | **87M** | **55.1** | 80.4 | **44.5** |

**Models and baselines.** We use PolyViT-Base and take linear probes of all the PolyViT models from Sec. 5.3, i.e. three single-modality models and two multi-modal models trained on all tasks with $L_{adapt} = 0$ and $L_{adapt} = L/2$ respectively. Our baseline models are those not performing co-training. Namely, we use ViT trained only on ImageNet-21K (ViT-Im21K) as a baseline, followed by ViT, ViViT and AST initialized from ViT-Im21K and finetuned on ImageNet, Moments in Time and VGGSound respectively (since these are the largest datasets for each respective modality).

**Discussion.** Table 3 shows how PolyViT trained on multiple modalities learns cross-modal feature representations that perform well on all 11 linear evaluation tasks across three different modalities (last two rows). This holds even when all the layers of the PolyViT transformer layer are shared, and thus the total number of parameters is roughly equal to a single-task model. PolyViT where the first half of the transformer encoder layers are modality-specific (final row), has more parameters and in general performs better. Furthermore, for the Epic Kitchens (video), Something-Something v2 (video) and Caltech 101 (image) datasets, multi-modal PolyViT transfers better than single-modality baselines. Table 3 thus demonstrates how co-training on multiple modalities facilitates learning powerful, transferable feature representations that can be used on multiple downstream tasks.

Models trained on only a single modality, as expected, do not in general learn feature representations that transfer well to other modalities. In particular, models trained on audio tasks do not transfer at all to images and videos, and vice versa. Models trained on video, however, still perform well on images, with video-trained models performing the best on the DmLab, PCAM and KITTI-Distance datasets. We believe this is due to the commonalities between the image and video modalities. Observe that in the majority of cases, single-modality PolyViT models perform better on linear probing than the corresponding single-task baselines, especially for video and audio.

### 5.5 Improving accuracy with single-modality co-training

Motivated by the performance of single-modality co-training in Tab. 2, we perform larger-scale co-training experiments with this method on audio and video classification. Tables 4 and 5 show that we achieve significant improvements in both of these domains whilst also using substantially fewer parameteters.

**Audio classification.** We compare to the current state-of-the-art using audio information only, MBT (Nagrani et al., 2021), using the same Base backbone, training on the balanced AS-500k subset, and other experimental settings as the authors (Nagrani et al., 2021). As shown in Tab. 4, we surpass the state-of-the-art on both datasets (AudioSet and VGGSound), whilst using about half the total parameters. We observe larger improvements (2.8%) on VGGSound, the smaller dataset. This is line with our findings from Sec. 5.3 and shows that co-training has a regularizing effect that reduces overfitting and improves performance the most on smaller datasets.

The second and third rows of Tab. 4 also show that training MBT on AudioSet and then finetuning on VGGSound, or vice versa, produces worse results than our co-training method. This shows that the regu-

Table 5: Comparison to ViVIT (Arnab et al., 2021) using the same experimental settings as Arnab et al. (2021). K400 and K600 denote Kinetics-400 and -600 respectively. MiT denotes the Moments in Time dataset.

| Model | #Models | #Params | K400 | | K600 | | MiT | |
|---|---|---|---|---|---|---|---|---|
| | | | Top 1 | Top 5 | Top 1 | Top 5 | Top 1 | Top 5 |
| ViViT | 3 | 913M | 80.6 | 94.7 | 82.5 | **95.6** | 38.0 | 64.9 |
| PolyViT | 1 | **308M** | **82.4** | **95.0** | **82.9** | 95.5 | **38.6** | **65.5** |

larization benefits of co-training are not solely from having access to more data than single-task baselines. As expected, finetuning MBT on the target dataset causes accuracy to degrade on the original dataset.

**Video classification.** We co-train PolyViT-Large with a smaller tubelet size (and hence greater number of tokens) of $2 \times 16 \times 16$ on Kinetics-400, -600 and Moments in Time. We compare to ViViT (Arnab et al., 2021) which a leading approach on this task, and directly comparable to our work as it uses the same architecture, initialization and number of tokens. As shown in Tab. 5, we surpass ViViT on all three datasets. Once again, the largest improvement of 1.8% is on Kinetics 400, which is also the smallest dataset, as co-training has a regularizing effect. Moreover, by co-training on three datasets, we reduce the total number of parameters required by almost three compared to separately trained ViViT models. Appendix E compares our models to other previous works on these audio and video datasets.

## 6 Conclusion and Future Work

By co-training PolyViT on a single modality, we have achieved signficant accuracy improvements on three video and two audio datasets, while reducing the total number of parameters linearly compared to single-task models. PolyViT co-trained on multiple modalities is even more parameter-efficient, still competitive with the state-of-the-art, and learns feature representations that generalize across multiple modalities. This enables us to learn new tasks by simply learning an additional output head. Co-training is simple and practical, as we do not need to tune hyperparameters on the joint space of all datasets, but can simply re-use training hyperparameters from single-task models. Moreover, we can achieve accuracy improvements from training for the same number of total steps. Currently we do not co-train on large-scale upstream datasets such as ImageNet-21K (Deng et al., 2009) and C4 (Raffel et al., 2019), and we do not handle the text modality either. We aim to explore this, and co-training with the text modality, in future work. An initial exploration of this idea is in Appendix F. Besides, we have only considered classification tasks, and not more complex tasks like object detection or captioning, and aim to explore this direction, e.g. bringing OWL-ViT (Minderer et al., 2022) to the ensemble of PoyViT, in future work.

### Broader Impact

Our work presents a method for performing image-, audio- and video-classification with a single parameter-efficient model. Classification of perceptual data (images, audio and video) is a general technology with a wide range of potential applications. While we are unaware of all potential applications, it is important to be aware that each application has its own merits and societal implications depending on the intentions of the individuals building and using the system. We also note that training datasets contain biases that may render models trained on them unsuitable for certain applications. It is possible that people use classification models (intentionally or not) to make decisions that impact different groups in society differently.

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

## Appendix

Appendix A contains additional details about our experimental settings, providing more information to Section 5.1 of the main paper. Appendix B provides more details for the experiments in Section 5.2 of the main paper. Appendix C shows further experimental details and results corresponding to Section 5.3 of the main paper. Appendix D provides additional details about the experiments in Section 5.4 of the paper. Finally. Appendix E provides additional experimental details and results corresponding to Section 5.5 of the main paper.

## A    Experimental set-up: additional details

**Task details and input dimensions.**    See Tables 6 and 7. For each task, the number of linear warmup steps is set as reported in (Dosovitskiy et al., 2021; Arnab et al., 2021; Nagrani et al., 2021). When co-training, we simply use the sum of all warmup steps for each co-trained task. We use a single momentum state when co-training, i.e. we don't maintain separate momentum states for each task or modality. Similar to Dosovitskiy et al. (2021), we select the best learning rate on a set $\{0.03, 0.1, 0.3\}$ using the validation score. For video and audio datasets, we reuse learning rates reported in Arnab et al. (2021) and Nagrani et al. (2021) respectively. As in (Arnab et al., 2021; Nagrani et al., 2021), we use zero initialization for output head kernels $\mathbf{W}_{out}$. For image datasets, on a single-task evaluation, we find that the LeCun normal $\mathbf{W}_{out}$ initializer Klambauer et al. (2017) works best. For all tasks, we perform gradient clipping with a maximum gradient norm of 1, as this was also used by (Dosovitskiy et al., 2021; Arnab et al., 2021; Nagrani et al., 2021).

For the "ViT-Im21K linear probe" baseline, we use the same training procedure as for single-task baselines, with the difference that 1) only the head parameters are updated and 2) on image tasks, we run separate learning rate grid searches on the set $\{0.03, 0.1, 0.3\}$.

**Train, validation and test splits.**    Similarly to Dosovitskiy et al. (2021), we take 2% of CIFAR 10/100 train sets for validation, 10% of Pets train set for validation and 1% of ImageNet-1k train set for validation. We use standard test sets for these datasets. For RESISC45, we use 20% of the train set for validation and 20% for testing. We use standard train, validation and test sets for video and audio tasks.

**Augmentation and regularization.**    We don't use augmentation for image tasks. We do video and audio preprocessing and augmentation as done in Arnab et al. (2021); Nagrani et al. (2021) respectively. For audio tasks, as in (Nagrani et al., 2021), we use Mixup (Zhang et al., 2017) with $\alpha = 0.3$ and stochastic depth regularization (Huang et al., 2016) with $p = 0.3$. Stochastic depth is applied along both audio adaptor and shared layers.

## B    Selecting the best task sampling schedule: additional experimental details

For the accumulating schedule, we set learning rate to the smallest value across tasks (0.03). We draw a random task order for the Task-by-task schedule, which is as follows: C100 $\rightarrow$ MiT $\rightarrow$ K400 $\rightarrow$ MiniAS $\rightarrow$ VGG $\rightarrow$ Pets $\rightarrow$ C10 $\rightarrow$ Im1K $\rightarrow$ R45.

## C    Co-training with PolyViT: additional experimental details and results

**Evaluation on video and audio tasks.**    To get test performance on video and audio tasks, we perform multiple-crop evaluation as described in Arnab et al. (2021); Nagrani et al. (2021) for videos and audio respectively.

**Results for the Large configuration.**    See Table 8. Since Nagrani et al. (2021) don't report results on a Large configuration, for audio tasks we do an additional hyperparameter tuning for single-task baselines on validation sets. As a result, we use Mixup $\alpha = 0.5, 0.7$ for MiniAS and VGGSound respectively. Also, we

Table 6: Experimental set-up: tasks and their properties. For image tasks, the indicated learning rates are obtained by a grid search over $\{0.03, 0.1, 0.3\}$ on single-task baselines using the validation set accuracy. These values are used for single-task baselines and for PolyViT variants.

| Dataset | Abbre-viation | Moda-lity | Clas-ses | Train size | Train steps | Learning rate | Warmup steps | $\mathbf{W}_{out}$ init |
|---|---|---|---|---|---|---|---|---|
| CIFAR 100 | C100 | Image | 100 | 50.0K | 10K | 0.03 | 500 | LeCun normal |
| CIFAR 10 | C10 | Image | 10 | 50.0K | 10K | 0.03 | 500 | LeCun normal |
| Oxford-IIIT Pets | Pets | Image | 37 | 3.68K | 500 | 0.03 | 100 | LeCun normal |
| RESISC45 | R45 | Image | 45 | 31.5K | 2.5K | 0.1 | 200 | LeCun normal |
| ImageNet-1k | Im1K | Image | 1000 | 1.28M | 20K | 0.03 | 500 | LeCun normal |
| Kinetics 400 | K400 | Video | 400 | 215K | 100.7K (30 epochs) | 0.1 | 2.5 epochs | Zeros |
| Moments in Time | MiT | Video | 339 | 791K | 123.6K (10 epochs) | 0.25 | 2.5 epochs | Zeros |
| Mini-Audioset | MiniAS | Audio | 527 | 20.4K | 15.9K (50 epochs) | 0.5 | 2.5 epochs | Zeros |
| VGGSound | VGG | Audio | 309 | 172K | 135K (50 epochs) | 0.5 | 2.5 epochs | Zeros |

Table 7: Input dimensions for different modalities. Sequence length is computed as $1+[(T/t)\times](H/h)\times(W/w)$ (one class token and patch tokens). Note that for shared transformer layers, we reuse the same parameters for sequences of different lengths.

| Modality | Input size, $[T\times]H \times W$ | Patch size, $[t\times]h \times w$ | Sequence length | Batch size |
|---|---|---|---|---|
| Image (pretraining) | $224 \times 224$ | $16 \times 16$ | 197 | 4096 |
| Image | $384 \times 384$ | $16 \times 16$ | 577 | 512 |
| Video | $32 \times 224 \times 224$ | $4 \times 16 \times 16$ | 1569 | 64 |
| Audio (spectrogram) | $800 \times 128$ | $16 \times 16$ | 401 | 64 |

use 30 epochs for MiniAS instead of 50 for the Base model. In addition, we run separate learning rate grid searches for all image tasks, separately for single-task baselines and ViT-Im21K linear probes. We apply all mentioned hyperparameter changes, obtained for the single-task baselines, to all PolyViT runs. In all other aspects, Large set-up is the same as Base.

## D  Linear probes: additional experimental details

**Task details.**  See Table 9. For linear probes, we use the same input dimensions as reported in Table 7. For image tasks, we reuse the number of train and warmup steps from the RESISC45 task (Table 6). For video and audio tasks, we used hyperparameters reported in (Arnab et al., 2021) and (Nagrani et al., 2021) respectively, with the difference that we only optimize output head parameters during training. As for the co-training setup, we use multiple-crop evaluation on video and audio tasks.

**Train, validation and test splits.**  For image tasks, we use 2% of the train set as a validation set and standard test sets. We use standard train, validation and test sets for video and audio tasks.

Table 8: Co-training with PolyViT, Large model configuration. Test accuracy (%) and mAP (for MiniAS, %) are reported. As indicated by the "# models" column, some rows correspond to multiple models, then the total number of parameters is computed across all models. Color bar: (lowest) ▨▨▨▨ (highest).

| Model | #Models | #Params | Image | | | | | Video | | Audio | |
|---|---|---|---|---|---|---|---|---|---|---|---|
| | | | C100 | C10 | Pets | R45 | Im1K | K400 | MiT | MiniAS | VGG |
| ViT-Im21k Linear probe | 1 | **312M** | 84.4 | 95.6 | 91.8 | 89.2 | 82.6 | 67.7 | 26.8 | 12.8 | 19.1 |
| Single-task baseline | 9 | 3033M | 93.3 | 99.2 | 94.8 | **97.3** | **85.1** | 79.6 | 37.1 | 30.0 | **51.8** |
| PolyViT, 1 modality | 3 | 917M | **93.9** | **99.4** | **95.5** | 96.9 | **85.1** | 80.6 | **38.8** | **37.9** | 50.7 |
| PolyViT, $L_{adapt}=0$ | 1 | **312M** | 91.4 | 99.0 | 94.7 | 96.8 | 82.6 | 78.9 | 35.8 | 33.3 | 49.9 |
| PolyViT, $L_{adapt}=L/2$ | 1 | 615M | 91.1 | 99.1 | 95.0 | 97.0 | 82.8 | **81.0** | 37.7 | 34.1 | 50.4 |

**Converting patch and positional embeddings for cross-modal probes.** In order to take linear probes of image-only models (ViT and PolyViT trained on images) on audio tasks (and vice versa), we leave patch embeddings as they are and 2D-interpolate positional embeddings to the correct resolution. When taking linear probes of video-only models on image or audio tasks, in order to obtain $16 \times 16$ patch embeddings, we take a sum along the first (frame) axis of 3D video patch embeddings of shape $4 \times 16 \times 16$. In order to adapt positional embeddings, we take a mean value of positional embeddings for each frame, and then 2D-interpolate the result to the correct resolution. When taking linear probes of image- or audio-only models on video tasks, we repeat 2D patch embeddings along the frame axis in order to obtain 3D patch embeddings. We also 2D-interpolate positional embeddings to the frame resolution and repeat them for each frame.

**Augmentation and regularization.** We don't use augmentation for image tasks. We do video and audio preprocessing and augmentation as done in (Arnab et al., 2021; Nagrani et al., 2021) respectively. As in (Arnab et al., 2021), we use Mixup (Zhang et al., 2017) with $\alpha = 0.3$ for the S-S v2 task.

Table 9: Tasks used for linear probes. Indicated learning rate grid search is done for all models using validation set performance.

| Dataset (task) | Abbre-viation | Moda-lity | Train steps | Learning rate | Warmup steps | $\mathbf{W}_{out}$ init |
|---|---|---|---|---|---|---|
| Caltech101 | C-ch101 | Image | 2.5K | Grid search, $\{0.03, 0.1, 0.3\}$ | 200 | LeCun normal |
| SUN397 | SUN397 | Image | 2.5K | Grid search, $\{0.03, 0.1, 0.3\}$ | 200 | LeCun normal |
| Dmlab | Dmlab | Image | 2.5K | Grid search, $\{0.03, 0.1, 0.3\}$ | 200 | LeCun normal |
| DTD | DTD | Image | 2.5K | Grid search, $\{0.03, 0.1, 0.3\}$ | 200 | LeCun normal |
| KITTI Distance | KITTI | Image | 2.5K | Grid search, $\{0.03, 0.1, 0.3\}$ | 200 | LeCun normal |
| PatchCamelyon | PCAM | Image | 2.5K | Grid search, $\{0.03, 0.1, 0.3\}$ | 200 | LeCun normal |
| Epic Kitchens | Epic K. | Video | 30 epochs | 0.5 | 2.5 epochs | Zeros |
| Something-Something v2 | S-S v2 | Video | 35 epochs | 0.4 | 2.5 epochs | Zeros |
| Kinetics 600 | K600 | Video | 30 epochs | 0.1 | 2.5 epochs | Zeros |
| Moments in Time (audio) | MiT-A | Audio | 10 epochs | 0.5 | 2.5 epochs | Zeros |
| Kinetics 400 (audio) | K400-A | Audio | 30 epochs | 0.5 | 2.5 epochs | Zeros |

Table 10: Extended comparison to ViViT (Arnab et al., 2021). The second row shows a ViVIT model, initialized from ImageNet-21K, and then finetuned on Moments in Time, Kinetics 600 and then Kinetics 400. This model has seen the same amount of training data as PolyViT, yet does not perform as well as PolyViT, showing that the improvements from co-training are not solely because PolyViT has access to more training data.

| Model | #Models | #Params | K400 | | K600 | | MiT | |
| | | | Top 1 | Top 5 | Top 1 | Top 5 | Top 1 | Top 5 |
|---|---|---|---|---|---|---|---|---|
| ViViT | 3 | 913M | 80.6 | 94.7 | 82.5 | **95.6** | 38.0 | 64.9 |
| ViViT: MiT $\rightarrow$ K600 $\rightarrow$ K400 | 1 | **308M** | 81.3 | 94.5 | 78.8 | 94.0 | 27.3 | 52.0 |
| PolyViT | 1 | **308M** | **82.4** | **95.0** | **82.9** | 95.5 | **38.6** | **65.5** |

## E State-of-the-art performance on one modality: additional experimental details and results

**Additional results on video datasets** Table 10 contains an extended comparison to ViViT (Arnab et al., 2021), compared to Table 5 of the main paper. The second row ("ViVIT: MiT $\rightarrow$ K600 $\rightarrow$ K400") shows our results when we train a ViViT model, by first finetuning an ImageNet-21K initialized model on Moments in Time, then Kinetics 600, and then finally Kinetics 400. This model has seen the same amount of training data as PolyViT, but performs worse on Kinetics 400 than PolyViT (PolyViT achieves 82.4, and ViViT achieves 81.3). This ViViT model, does however, still outperform a ViViT model finetuned solely on Kinetics 400 from ImageNet-21K initialization (first row). This result, like Table 4 of the main paper for audio, shows that the benefits of co-training are not only because the co-trained PolyViT model has access to more data.

For our additional baseline, ("ViVIT: MiT $\rightarrow$ K600 $\rightarrow$ K400"), we retain the output linear head for each class. Consequently, the accuracy for MiT and Kinetics 600 degrades as the model is trained on Kinetics 400. Note that Kinetics 600 is a superset of Kinetics 400, which is why the overall accuracy drop on Kinetics 600 is low. Furthermore, note that the goal of this paper is not to consider the "continual learning" (Kirkpatrick et al., 2017) problem, which aims to train a model on a new dataset, without losing performance on previous datasets the model was trained on.

**Detailed experimental settings** For the PolyViT experiment on the video modality, we reuse hyperparameters reported in Arnab et al. (2021) for Kinetics 400/600 and Moments in Time. See Table 12 for the dataset details and exact hyperparameters used during the experiment. These hyperparameters coincide with those reported in Table 6 for Kinetics 400 and Moments in Time and in Table 9 for Kinetics 600. The only difference is that we use a more granular 3D patch size ($2 \times 16 \times 16$) and Large model configuration.

For the audio experiment, similarly, we reuse all hyperparameters reported in Nagrani et al. (2021) for AS-500k and VGGSound experiments (audio-only). See Table 13 for the dataset details and exact hyperparameters used for the experiment. These hyperparameters almost coincide with those reported in Table 9 with a change MiniAS $\rightarrow$ AS-500k. The only exception is that we use 30 epochs and Mixup $\alpha = 0.5$ for AS-500k.

## F Upstream co-training of image and text

This section presents an initial experiment of "upstream" pre-training of image classification on JFT (Sun et al., 2017) and BERT-pretraining on Wikibooks (Devlin et al., 2019). Here, we train a PolyViT-Base model from random initialisation. In contrast to the experiments of the main paper which finetuned ImageNet-21K pretrained checkpoints, we cannot use pretrained models here due to the vast domain differences between the image and text modalities.

For JFT classification and BERT pretraining, we follow the experimental hyperparameters and protocols of the original authors (Dosovitskiy et al., 2021; Devlin et al., 2019).

Table 11: "Upstream finetuning" on JFT image classfication and BERT text-pretraining. Note that models are trained from random initialisation.

| | Image (JFT) | Text (Wikibooks) | |
| --- | --- | --- | --- |
| | Accuracy | MLM Accuracy | Next Sentence Prediction |
| ViT-Base | 48.0 | – | – |
| BERT-Base | – | 72.8 | 99.0 |
| PolyViT-Base | 50.0 | 70.9 | 97.9 |

As shown in Table 11, compared to a single-task baseline, PolyViT improves slightly on JFT classification, but degrades marginally on text classification metrics.

Note that in this experiment, we have not evaluated the representations learned by the network, for example, by finetuning on dowstrean tasks. We leave this, and further effors on "upstream co-training" for future work.

Table 12: Set-up for the co-training on videos. Train steps and warmup steps are summed to get the number of train and warmup steps during co-training as we use the "Weighted" task sampling method.

| Dataset | Moda-lity | Clas-ses | Train size | Train steps | Batch size | Learning rate | Warmup steps | $\mathbf{W}_{out}$ init |
| --- | --- | --- | --- | --- | --- | --- | --- | --- |
| Kinetics 400 | Video | 400 | 215K | 101K (30 epochs) | 64 | 0.1 | 2.5 epochs | Zeros |
| Kinetics 600 | Video | 600 | 363K | 170K (30 epochs) | 64 | 0.1 | 2.5 epochs | Zeros |
| Moments in Time | Video | 339 | 791K | 123.6K (10 epochs) | 64 | 0.25 | 2.5 epochs | Zeros |

Table 13: Set-up for the co-training on audio. Train steps and warmup steps are summed to get the number of train and warmup steps during co-training as we use the "Weighted" task sampling method.

| Dataset | Moda-lity | Clas-ses | Train size | Train steps | Mixup | Batch size | Learning rate | Warmup steps | $\mathbf{W}_{out}$ init |
| --- | --- | --- | --- | --- | --- | --- | --- | --- | --- |
| AS-500k | Audio | 527 | 509K | 239K (30 epochs) | 0.5 | 64 | 0.5 | 2.5 epochs | Zeros |
| VGGSound | Audio | 309 | 172K | 135K (50 epochs) | 0.3 | 64 | 0.5 | 2.5 epochs | Zeros |

