# OpenReview forum: "PolyViT: Co-training Vision Transformers on Images, Videos and Audio"
_TMLR — Accepted by TMLR_

### Review · Reviewer_iqC8 · 2022-09-12

**Summary Of Contributions:**

This work trains a single transformer model named PolyViT to perform multi-modal tasks. It proposes and compares various co-training procedures with different task sampling schedules. This work trains multi-modal PolyViT on 9 datasets across three modalities (of image, audio, and video) and outperforms state-of-the-art single-task baselines (MBT and ViVIT) on VGGSOund and Kinetecs-400/600.

**Broader Impact Concerns:**

No specific ethical concern in this work.

**Requested Changes:**

Please see the comments in Weaknesses.

My general feeling on this work is that it is decent but outdated by recent unified models (Unified IO, Perceived IO, and many more) in various aspects – technical novelty, model and data scales, and experiments.


**Strengths And Weaknesses:**

<Strengths>
1. This work shows that a single transformer model trained on multi-modal tasks can outperform single-task SOTA models.

2. It proposes multiple co-training schemes with different task sampling schedules – Task-by-task, accumulated, Alternating, Uniform, and Weighted.

<Weaknesses>
1. I think the main issue of this work is that it is somewhat outdated by recent progress of unified multimodal transformer models.
- For example, recent works like Unified-IO (https://unified-io.allenai.org/) by AI2 and Perceiver IO (https://www.deepmind.com/open-source/perceiver-io)  by DeepMind address the training of transformers in much more challenging multimodal settings – more data, more parameters, more modalities and more tasks.

2. The technical novelty is limited.
- Training a single transformer model with multimodal datasets is not new.
- Using different configurations of multimodal data in a minibatch is not new.
- The proposed PolyViT may seem to be a marginal straightforward upgrade of ViT  toward multimodal data with some heuristics.

3. Experiment evaluations are not convincing.
- This work mainly compares with ViT models as shown in Table 2-3.
- There is no comparison with SOTA unified multi-modal transformer models.

---

> ### Author Response · Authors · 2022-09-19
> **Author response**
>
> Thank you for your prompt review.
>
> Note that we already discussed Perceiver in the Related Work section of our paper, and we have now additionally discussed Perceiver-IO and Unified-IO in our updated manuscript.
>
> Perceiver-IO, like Perceiver, develops a transformer architecture that can process a wide range of modalities. Crucially, the authors train a *different* Perceiver / Perceiver-IO network for each separate task. The Perceiver line of works therefore do not address the problem of training a *single* network to perform multiple tasks whilst sharing almost all learned parameters, as done by our work.
>
> Unified-IO, like our work, does address multiple tasks across multiple modalities with a single network. However, we note that this paper only appeared recently on Arxiv and is thus concurrent work to ours. Moreover, the results of Unified-IO are not competitive with ours on standard tasks like ImageNet classification as shown below. The same is also true for Perceiver and Perceiver-IO which also achieve lower results with more computationally expensive models.
>
> |                                      | ImageNet Top-1 | GFLOPs   | Params (M) |
> |--------------------------------------|:--------------:|:--------:|:----------:|
> | Perceiver (2D Fourier)               | 78.6           | 404      | 42.1       |
> | Perceiver IO (2D Fourier)            | 79.0           | 407      | 48.4       |
> | Perceiver (conv)                     | 77.4           | 367      | 42.1       |
> | Perceiver IO (conv)                  | 82.1           | 369      | 48.6       |
> | Unified-IO Base                      | 63.3           | --       | 241        |
> | Unified-IO XL                        | 79.1           | --       | 2800       |
> | PolyViT-Base  | **83.1**       | **17.6** | 93         |
>
>
> Note that the Perceiver family of models use significantly higher FLOPs (numbers from Table 7 of their appendix). This is likely due to the fact that Perceiver treats each pixel as a token, whereas vision transformers embed patches into tokens. We could not add in FLOPs for Unified-IO at this time as they are not available in the paper, and the code has not been released yet.
>
> We will also compare to other works in the next revision if the reviewer provides additional references.
>
> Following the TMLR guidelines, we will upload our revised manuscript following the other reviews.

---

### Review · Reviewer_7MGN · 2022-09-19

**Summary Of Contributions:**

This paper studies training a single transformer model to process multiple modalities and datasets, whilst sharing almost all of its learnable parameters. For this purpose, a new transformer model called PolyViT was proposed.

By co-training on different tasks and modality, PolyViT achieves significant accuracy improvements on several video- and audio-classification datasets. The obtained parameter-efficient model generalizes across multiple domains outperforming a state-of-the-art single-task baseline on 2 of these datasets, whilst achieving competitive performance on the others.


**Broader Impact Concerns:**

Broader impact concerns were briefly discussed.

**Requested Changes:**


Could you please explain what you mean by "interpolated from the pretrained model" in the positional embeddings, p, which depend on the sequence length N (and thus the input resolution), are interpolated from the pretrained model to the finetuned model?

How do you accumulate gradients in different sampling schedules? Do you just take the average?

Training phases should be further analyzed. For instance, have you observed any vanishing/exploding gradients, why adaptor layers do not help, why does accuracy drop too much in some cases in Table 3 etc.?

What is the array referred in "by randomly permuting an array with U elements"?

Could you please further explore the claim "Models trained on only a single modality, as expected, do not in general learn feature representations that transfer well to other modalities. In particular, models trained on audio tasks do not transfer at all to images and videos, and vice versa"?

How do you train modality-specific transformer layers (denoted as modality adaptor layers)? This part was not explained clearly.

Have you used other optimization methods for training? Is there any particular reason for choosing SGD? An ablation of optimizers would be useful.



**Strengths And Weaknesses:**

The paper is well written in general. Initial analyses show that the proposed method outperforms a state-of-the-art single-task baseline on 2 of these datasets, whilst achieving competitive performance on the others.

There are several unclear statements in the paper, some of which are discussed below.

A major weakness of the paper is the lack of detailed analyses. The current paper proposes a black box method to train a multi-modal transformer with parameter sharing, which achieves good accuracy. To improve the paper, detailed analyses of its underlying mechanism and several ablations should be provided.

---

### Review · Reviewer_nHdK · 2022-09-21

**Summary Of Contributions:**

The article presents PolyViT, a single transformer model that can address multiple tasks in multiple modalities. The model uses modality-specific tokenizers and task-specific heads. During training, the model sees data of the multiple modalities and tasks, with the manuscript analyzing different training choices (e.g. task order, sampling, etc.). Results show that the approach achieves results comparable to models trained for the specific modalities and tasks.

**Broader Impact Concerns:**

None but the broader impact section of the paper perfectly details potential misuses of the users and biases on the datasets.

**Requested Changes:**

Following on the weaknesses described above, I would recommend the following revisions (ordered per importance):
1. The manuscript should provide a more thorough comparison with previous works (MultiModel in particular). This holds both for quantitative comparisons (i.e. either using the same setup or adapting the model to the considered benchmarks) as well as discussions on the limitations of PolyViT (i.e. focus on classifications).
1. Are there reasons why PolyViT cannot be readily applied to text? Even including negative results would be helpful to add more insights to the work.
1. Analyze how PolyViT performs if trained by varying the number of tasks per modality and on extreme setups (e.g. 1 task per modality).
1. (minor) The abstract should describe PolyViT.
1. (minor) Table 4, for VGGSound top-5, the bold is misplaced.

**Strengths And Weaknesses:**

**Strengths**:
1. The approach is very simple, i.e. specific tokenizers for each modality and a specific head for each task. It is easy to implement and can constitute a solid baseline for future work.
2. The results show that the model is very effective: Table 2 shows how PolyViT achieves comparable performance to single-task baselines while using much fewer parameters. These results are confirmed in the linear probing case (Table 3), showing the generalization capabilities of the features learned by the model.
3. Table 1 provides a thorough overview of the different training choices and how they impact the final results. These clearly show that taking into account the size of each dataset and training the model for all tasks at once is fundamental for achieving good results.
4. The writing is clear and easy to follow.


**Weaknesses**:
1. The model is not compared with any baseline on multi-domain/task learning. Arguably, the main competitors is MultiModel (Kaiser et al. 2017). The introduction states that unlike (Kaiser et al. 2017), PolyViT achieves results comparable to the state of the art. However, the two models are compared on different tasks and modalities (e.g. the former image-text-audio-categorical, the latter image-video-audio), with (Kaiser et al. 2017) focus their analyses on tasks beyond classification (e.g. captioning, translation). To support the claim that PolyViT is better than MultiModel, the two should be compared on the same ground. Note also that the paper should describe the shortcomings of the model (tested only on classification) w.r.t. to MultiModel and other previous works (e.g. Hu and Singh 2021) tested on more complex tasks.

1. Since the model can already be applied to text, showing how PolyViT works on this would have made the article more complete. In case PolyViT struggles with text, the manuscript should report and discuss such findings.

1. From the experiments it is unclear how the performance of PolyViT changes w.r.t. the number of tasks per modality. For instance, if the model is trained on one task per modality, the performance would still benefit from co-training? Analyzing this dependence would add more insights into the effectiveness of the PolyViT framework and possible directions for future works. Note also that Table 3 shows unclear trends in this regard (i.e. audio does not benefit from datasets of other modalities while images and video ones achieve comparable performance).

1. While the article is well-written, the abstract does not describe how PolyViT works (i.e. modality-specific tokenizers, task-specific heads).

1. As a final, minor note: it would have been interesting to compare the approach with other common multi-domain learning strategies (e.g. the trend initiated by Rebuffi et al. 2017) to show the extent to which co-training is sufficient in this multimodal setup. This would not only add baselines to the comparisons but also more insights on whether it is important to refine part of the backbone (beyond the *L_adapt* variant) in this multimodal setup.

---

### Decision · Action_Editors · 2022-11-01

**Recommendation:** Accept with minor revision

**Comment:**

This paper is reviewed by three experts on this topic. All the reviewers give very good comments and suggestions. The authours provide extensive and insightful discussion, as well as some additional experiments. Most of concerns on the paper are addressed. Although two reviewers have still concern about the novelty of this model and are thus slightly negative to the paper, the AE suggested the acceptance, due to several reasons.
(1) Overall, the paper is well written, and clean. The experiments are well supporting the claims and contributions.
(2) A single simple and effective model to unify several datasets from different tasks/domains is truly valuable to the community in general.
(3) Some engineering techniques are useful and of good value to the audience.

I would suggest the accept with minor, in order to give the authous the final chance of proof-reading and update the manuscript,  inserting the code links and so on. BTW, is the word 'Poly' short for something, or just using the meaning of the word here?



**Audience:**

This work is valuable to the researchers of video and image understanding, and audio signal processing. It works in a promising way of unifying several different sub-communities, as it gives a single, effective and unified transformer model.

**Claims And Evidence:**

This paper presents a PolyViT model, a single transformer model that can address multiple tasks in multiple modalities.
The model uses modality-specific tokenizers and task-specific heads.  The overall architecture is very simple, clean and effective. Some   engineering techniques like initialization, co-training strategy, optimization make the model work well. The results are convincing with large-scale experiments to support the claims.